# Loss of transcriptional heterogeneity in aged human muscle stem cells

**Emilie Barruet[1,2]***, **Katharine Striedinger[1]**, **Pauline Marangoni[2]**, **Jason H. Pomerantz**[ID][1]*

**1** Departments of Surgery and Orofacial Sciences, Division of Plastic and Reconstructive Surgery, Program in Craniofacial Biology, Eli and Edythe Broad Center of Regeneration Medicine, University of California San Francisco, San Francisco, California, United States of America, **2** Program in Craniofacial Biology and Department of Orofacial Sciences, University of California, San Francisco, California, United States of America

* Jason.Pomerantz@ucsf.edu (JHP); Emilie.Barruet@ucsf.edu (EB)

## Abstract

Age-related loss of muscle mass and function negatively impacts healthspan and lifespan. Satellite cells function as muscle stem cells in muscle maintenance and regeneration by self-renewal, activation, proliferation and differentiation. These processes are perturbed in aging at the stem cell population level, contributing to muscle loss. However, how representation of subpopulations within the human satellite cell pool change during aging remains poorly understood. We previously reported a comprehensive baseline of human satellite cell (Hu-MuSCs) transcriptional activity in muscle homeostasis describing functional heterogenous human satellite cell subpopulations such as CAV1+ Hu-MUSCs. Here, we sequenced additional satellite cells from new healthy donors and performed extended transcriptomic analyses with regard to aging. We found an age-related loss of global transcriptomic heterogeneity and identified new markers (*CAV1*, *CXCL14*, *GPX3*) along with previously described ones (*FN1*, *ITGB1*, *SPRY1*) that are altered during aging in human satellite cells. These findings describe new transcriptomic changes that occur during aging in human satellite cells and provide a foundation for understanding functional impact.

## Introduction

Aging in skeletal muscle is characterized by a decline in muscle mass and regenerative capacity manifested in humans by decreased muscle strength and slow healing after injury [1,2]. At the cellular level, muscle fiber growth, turnover and regeneration are driven by muscle stem cells also known as satellite cells [3], characterized by the expression of PAX7. Satellite cells undergo activation and proliferation upon injury, and then either differentiate to generate new muscle fibers or return to a quiescent state to reconstitute the satellite cell pool. Therefore alterations in satellite cells during aging could underlie associated changes in muscle bulk and function.

There has been discordance in the literature with several studies reporting age-related loss in number and function of satellite cells [4–6] while others have reported no significant reduction or change [7,8]. We observed a modest decrease in satellite cell number in samples from

**Data Availability Statement:** Single cell gene expression fastq files and filtered matrices have been deposited (GSE196554). De-tailed scripts can be found here, https://github.com/EmilieB12/Aging_Hu-MuSCs.

**Funding:** This work was supported by NIH R01AR072638-03 to JHP The funders had no role in study design, data collection and analysis, decision to publish, or preparation of the manuscript.

**Competing interests:** The authors have declared that no competing interests exist.

elderly (>81 years) human individuals [9]. Age can also cause intrinsic changes of satellites cells, their niche or both [5,10–14]. Recent advances in single-cell genomics have allowed the discovery of novel aspects of aging in different tissues, which includes changes in cell heterogeneity, distribution of cellular states and gene expression levels [15–18]. Studies in mice profiling the transcriptome of muscle stem cells along differentiation pathways have revealed age-related changes such as a decrease in expression of extracellular matrix (ECM), migration and adhesion genes [19]. Most prior intrinsic satellite cell aging studies have been performed in mice, with few efforts to translate those findings to humans [20]. Thus, a comprehensive characterization of the impact of aging on human satellite cells is still lacking.

We previously demonstrated that under homeostatic conditions, human satellite cells are transcriptionally heterogeneous, which enabled us to separate functionally distinct human satellite cell subpopulations [21]. Therefore, a comprehensive understanding of how the repartition of these subpopulations and their gene expression vary along with aging is now feasible. In this study, we used single cell RNA sequencing of satellite cells from additional human muscle samples to further analyze existing datasets with regard to aging. We demonstrate that there is a loss of transcriptional heterogeneity with aging and identify new genes that are differentially expressed during aging.

## Materials and methods

### Human specimen procurement and Hu-MuSCs isolation

This study was conducted under the approval of the Institutional Review Board at The University of California San Francisco (UCSF). Biopsies were obtained from individuals undergoing surgery at UCSF. Written informed consent was obtained from all subjects. All types of muscle used for each experiment are listed in S1 Table. Additional CXCR4+/CD29+/CD65+ Hu-MuSC samples were isolated as described in [9,21–23].

### Single cell RNA sequencing and analysis

Single cell RNA sequencing and gene core matrices retrieval of additional samples were performed as described in [21]. Gene-barcoded matrices were analyzed with the Python package Scanpy version 1.9 (RRID:SCR_018139) [24]. For each sample, cells with fewer than 500 genes, greater than 7000 genes and genes expressed in fewer than 5 cells were not included in the downstream analyses. Cells with more than 15% mitochondrial counts were filtered out. Each sample was first merged into its own age group with batch balanced k nearest neighbor (BBKNN) algorithm [25] to remove potential technical variation between samples. A resolution of 0.5 was used for all subset age group. Cluster were annotated using known markers found in the literature combined with differentially expressed genes (Wilcoxon test, function sc.tl.rank_genes_groups). Since after filtering the adult group contained 66,905 cells, and the young and aged group contained 11,502 cells and 9,407 cells respectively, to avoid cofounding factors due to discrepancy in cell number among the three age groups, we downsampled the adult cell group to 11,000 cells using the sc.pp.subsample function, which is a state of art to eliminate such bias [26]. Following this, the 'adult' and 'aged' datasets were integrated onto the annotated 'young' dataset using the Scanpy INGEST function sc.tl.ingest. Differential expression analysis was performed between the age groups using the same Wilcoxon statistical test, as implemented in Scanpy. Marker gene expression was visualized using either dot-plots, where the size of the dot reflected the percentage of cells expressing the gene and the color indicated the relative expression, or violin plots, with the width of the violin plot depicting the larger probability density of cells expressing each gene at the indicated expression levels.

### scVelo and PAGA trajectory analysis

Count matrices (unspliced) and mature (spliced) abundances were generated for each sample from fastq files using Kallisto/loompy 3.0 package (RRID:SCR_016666). scVelo v0.2.4 package (RRID:SCR_018168) implemented into Scanpy was used to perform RNA velocity analysis [27]. Datasets were processed using the recommended parameters as described in Scanpy scVelo implementation [27]. The age group samples were pre-processed using scv.pp.filter and scv.pp.normalize followed by scv.pp.moments functions for detection of minimum number of counts, filtering and normalization. scv.tl.velocity and scv.tl.velocity_graph functions were used to calculate and visualized gene specific velocities. Gene ranking for each age group resulting from differential velocity t-test was perform using the scv.tl.rank_velocity_genes. Scanpy implemented partition-based graph abstraction (PAGA) functions (scv.tl.paga and scv.pl.paga) was used to assess the data topology with weighted edges corresponding to the connectivity between two clusters. Default parameters were used [28].

### Gene ontology analysis

Differentially expressed genes, p-values and fold changes were used as input to generate GO-term enrichment with the clusterProfiler package in R. Thresholds were set p-value <0.05 and fold change >1 for the GO-term analysis.

### Data and materials availability

Single cell gene expression fastq files and filtered matrices have been deposited (GSE196554). Detailed scripts can be found here, https://github.com/EmilieB12/Aging_Hu-MuSCs.

## Results

### Decreased transcriptional heterogeneity in Hu-MuSCs during aging

Our previous work identified functionally heterogenous human satellite cell subpopulations. We asked whether the distribution of those subpopulations or their transcriptome are altered in aging. We performed single cell RNA sequencing of highly purified human satellite cells from new healthy donors (S1 Table) and pooled them with our previous dataset [21]. Our analysis workflow is described in Fig 1. The single cell sequencing data for each sample were analyzed using SCANPY [24], merged into their respective age group using the BBKNN (batch balanced k nearest neighbors) integration algorithm [25] to remove batch effect, and visualized in uniform manifold approximation and projection (UMAP) graphs (S1A and S1B Fig). Samples were distributed in 3 age groups: young [<30 y.o.], adult [35–66 y.o.] and aged [>70 y.o.]. All samples within each group were pooled. We identified 12, 9 and 10 clusters for the young, adult and aged groups, respectively. Myogenic, cycling and stemness genes were expressed as previously described [21] in each age group. Moreover, similar cluster markers such as AP-1 transcription factor unit (*JUN*, *FOS*), *COL1A1* (Collagen Type Alpha 1 Chain), *SOX8* (Sry-Box Transcription Factor 8), *IGFBP7* (Insulin Like Growth Factor Binding Protein 7), *MX1*, *HSPA1A* (Heat Shock Protein A, Hsp70)) or *CAV1* (Calveolin-1) were found in each age group (S1B and S1C Fig). To compare each age group to another, we used INGEST [29,30]. Unlike BBKNN or CCA (Canonical Correlation Analysis, e.g. in Seurat) where datasets are integrated in a symmetric way, INGEST integrates asymmetric datasets into a 'reference' annotated dataset. We found that the clusters obtained in our grouped young samples were robustly defined by accepted markers, in addition to having the greatest transcriptional variability (estimated through a higher number of clusters). Hence, we used this young group as our 'reference' annotated dataset to best identify potential differences in the transcriptional signatures

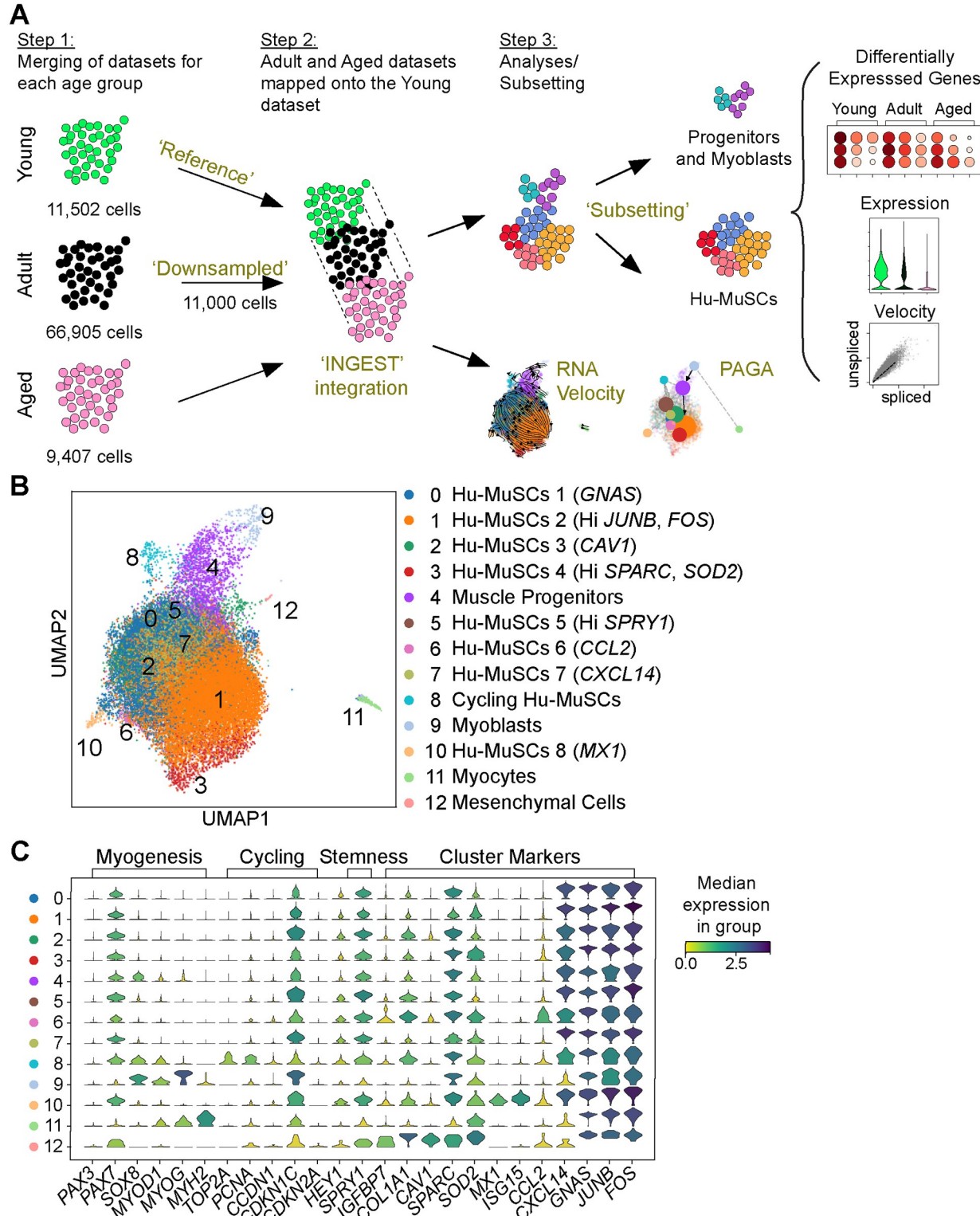

**Fig 1. Analysis workflow and scRNA sequencing clusters.** (A) Schematic of the analysis. (B) UMAP of merged age groups using INGEST with labeled clusters. (C) Violin plots displaying the expression of myogenic, cycling, stemness and cluster marker genes for each cluster.

induced by aging (S2A and S2B Fig). This approach allowed us to detect the biological variation observed with aging.

Prior to mapping, the adult group of cells was downsampled to 11,000 cells to remove co-founder effect resulting from differences in cell number among the 3 age groups. Among the 12 clusters, clusters 0–3, 5–7 and 10 consisted of quiescent Hu-MuSCs while cluster 4, 8, 9, 11 and 12 consisted of muscle progenitors, cycling Hu-MuSCs, myoblasts, myocytes and mesenchymal cells, respectively. We confirmed that each Hu-MuSC cluster was found to have a unique transcriptomic fingerprint. Cluster 0 was characterized by the upregulation of *GNAS* and cluster 1 by the upregulation of *JUNB* and *FOS*. Cluster 2 contained the *CAV1* expression cells while cells expressing high levels of *SPRY1* (Sprouty RTK Signaling Antagonist 1) were found in cluster 5. Cluster 6 and 7 consisted of cells expressing cytokines *CCL2* and *CXCL14*. Finally, the recently described [31] *MX1* satellite cell subpopulation was identified as cluster 10 (Fig 1B and 1C). The low levels of *PAX3* transcripts found in cluster 10 reflects known limitations of single cell RNA sequencing. In addition, the expression of PAX3 in adult human stem cells has yet to be fully elucidated [32].

The INGEST integration allowed us to compare the distribution of the young, adult and aged Hu-MuSCs among the different clusters. A UMAP based-density plot revealed a decrease of cluster coverage with aging (Fig 2A and S2B Fig). The majority of aged Hu-MuSCs were located in cluster 1 (65%, Hi *JUNB*, *FOS*) and cluster 7 (10.9%, *CXCL14*), while young Hu-MuSCs were distributed among a larger number of clusters such as cluster 2 (14.8%,*CAV1*), cluster 3 (13.3%, Hi *SPARC*, (Secreted Protein Acidic and Cysteine Rich, an ECM protein [33]), *SOD2* (Superoxide Dismutase 2)), cluster 5 (3.9% Hi *SPRY1*) and cluster 6 (2.4%, *CCL2*). Adult cells were predominantly present in cluster 0 (33.5%), 1 (48.9%) and 7 (5.1%) (Fig 2A and 2B, S2B and S2C Fig). Thus, distribution of cells per cluster varies with aging and there is a relative loss of transcriptional heterogeneity in aged Hu-MuSCs.

Since the distribution of cells per cluster appears affected by aging, we aimed to investigate the direction and speed of movement of cells in clusters inferred by RNA velocities [34]. To understand the cellular dynamics of Hu-MuSCs and population kinetics during aging, we applied the scVelo and partition-based graph abstraction (PAGA) trajectory algorithm [27]. scVelo and PAGA analyses suggest that in young cells, cluster directionality is heterogenous, contrary to adult and aged cells where cell states appear to commit to cluster 1 (Fig 2C and 2D). Since the weighted edges correspond to the connectivity between two clusters in the PAGA analysis, we were able to more closely investigate the connectivity between clusters. In the three age groups we found a strong connection axis between clusters 1 (Hu-MuSCs, Hi *JUNB*, *FOS*), 4 (Muscle Progenitors) and 9 (Myoblasts) with directional kinetics toward the less differentiated states. Aged cells from other Hu-MuSC clusters (0, 2, 3, 6, 10) converged toward cluster 1 while in the young age group, Hu-MuSC clusters appear to be in a non-convergent steady state. We also found that cells from cluster 7 (*CXCL14*) in the young group were not connected to any other cluster in contrast to adult and aged cells (Fig 2D). These findings suggest that aging Hu-MuSCs have a decrease in transcriptional heterogeneity characterized by cluster specificity and convergent RNA velocity.

## Extracellular-matrix and adhesion gene expression decreases with aging in Hu-MuSCs

Since we observed a loss in transcriptional heterogeneity, we asked which genes may be differentially expressed during aging. We excluded activated, differentiated and non-myogenic cells and focused on the Hu-MuSC clusters (0–3, 5–7 and 10) solely to assess modulations in transcriptional signatures and highlight age-related modifications (Fig 3A). We then identified

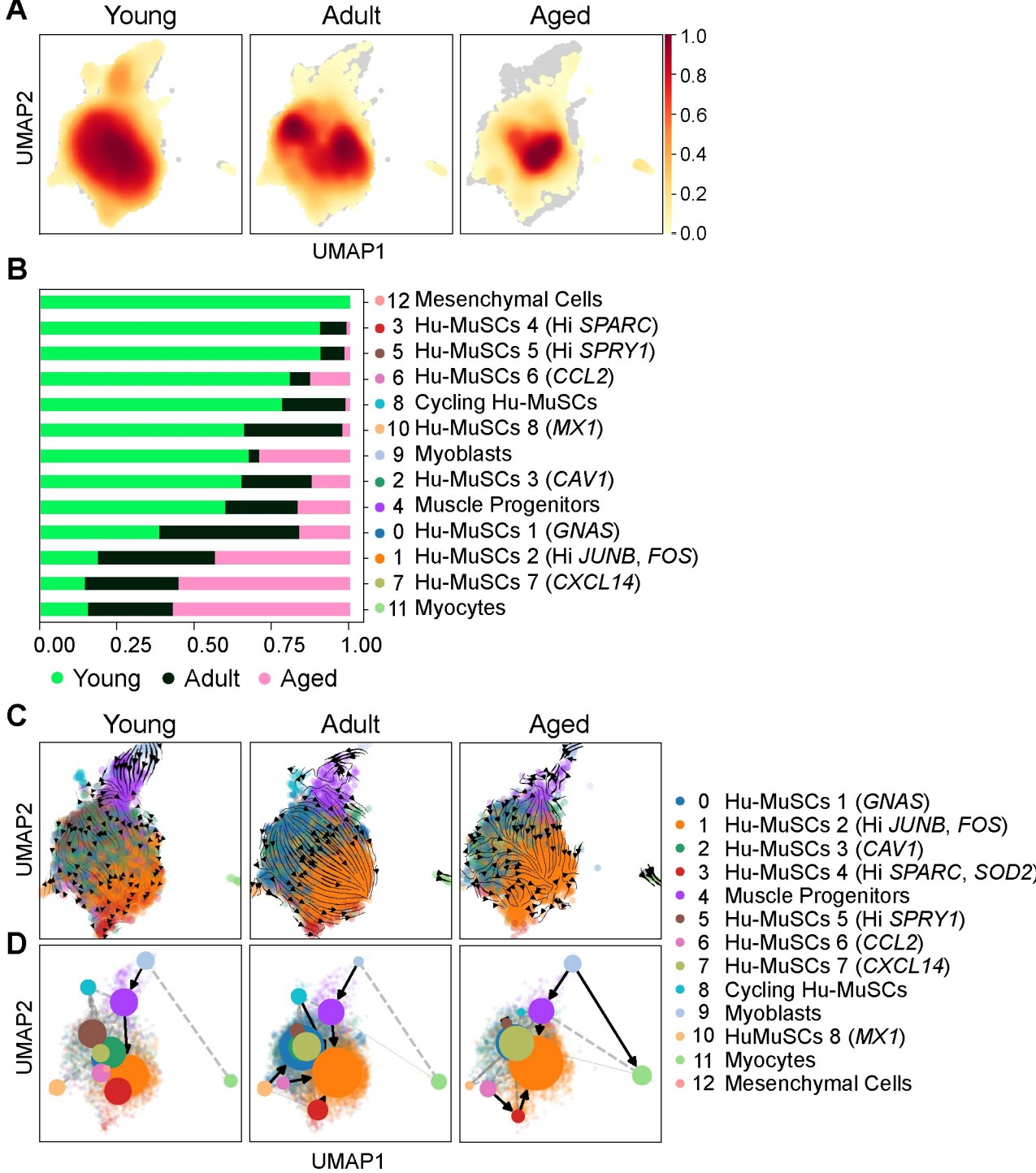

**Fig 2. Distribution of Hu-MuSC subpopulations during adult aging.** (A) UMAP density plot for each age group (Young, Adult, Aged). (B) Proportion plot of cells assigned to each age group according to each cluster. (C) RNA velocities projected onto the UMAP clusters for each age group. (D) PAGA analysis for the different age group. Weighted edges correspond to the connectivity between two clusters.

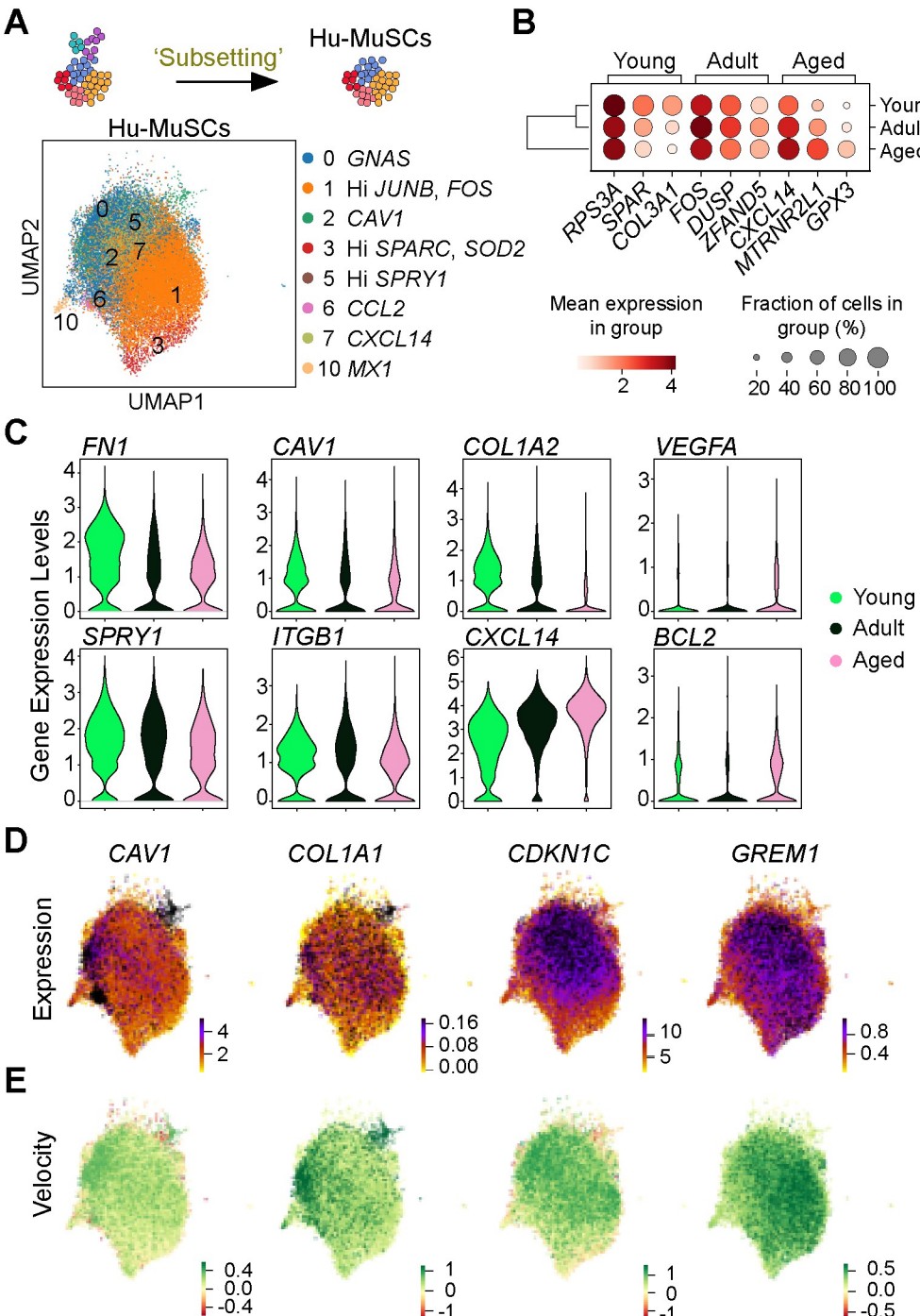

**Fig 3. Gene expression and velocity analyses of Hu-MuSCs of young, adult and aged groups.** (A) Schematic of the sub-clustering and UMAP of merged age groups for only the human muscle stem cells (cluster 0–3, 5–7 and 10). Only Hu-MuSCs were analyzed. (B) Dot plot displaying the top 3 genes differentially expressed for each age group in the whole Hu-MuSCs subset. (C) Violin plots of the expression of genes altered with aging. (D) Expression of relevant genes that inferred age-specific differential velocity. (E) Velocity of genes from (D).

differentially expressed genes for the three age groups, where each age group was compared to one another. Notably we found collagen genes (i.e. *COL3A1*) to be significantly expressed in young Hu-MuSCs, while *DUSP1* (can inactivate MAPK proteins and has been reported to increase upon Hu-MuSC activation in culture [35]) and *ZFAND5* (a proteosome activator [36]) were significantly expressed in adult Hu-MuSCs, and *CXCL14* and *GPX3* (Glutathione Peroxidase 3, a retinoid-responsive gene [37]) in aged Hu-MuSCs (Fig 3B). These results also corroborate previously described genes affected by aging such as *FN1* (Fibronectin 1), *ITGB1* (Integrin Subunit Beta-1), *SPRY1* [13,14,38] which were decreased with aging. Moreover, expression of other ECM and adhesion related genes (*CAV1*, *COL1A2*) decreased in aged cells. The expression of mTor pathway target genes such as *BCL2* and *VEGFA* (Vascular Endothelial Growth Factor A) was increased in adult and aged Hu-MuSCs (Fig 3C).

Since the distribution of cells among the different clusters as well as population kinetics changed with aging, we asked if the inferred directions described in Fig 2C were supported by any specific genes or transcriptional program. In comparing the three age groups, we found that genes including *CAV1*, *COL1A1*, *CDKN1C* (Cyclin Dependent Kinase Inhibitor 1C, regulates cell proliferation [39]) and *GREM1* (Gremlin1, a BMP antagonist [40]) had age-specific differential velocity expression meaning that those genes were transcribed at significantly higher or lower levels compared to their age group counterpart (Fig 3D and 3E). The UMAP of velocity expression showed that *CAV1* and *COL1A1* had increased velocity in the clusters enriched in the young samples (cluster 2, 3 and 5) while *CDKN1C* and *GREM1* displayed increased velocity in clusters enriched with the adult (cluster 0) and aged (cluster 1) samples, respectively (Fig 3E). We also found additional genes with an age-specific significant differential velocity. These include *DIO2* (Type 2 Iodothyronine Deidinase, coverts thyroid prohormone [41]), *EDN3* (Endothelin-3, mediates the release of vasodilators [42]), *NPTX2* (Neuronal Pentraxin 2, which affect tumor progression [43]), *KLF6* (Krueppel-like Factor 6, a tumor suppressor [44]), *LPL* (Lipoprotein Lipase, involved in lip metabolism [45]) and *MAP1B* (Microtubule Associated Protein [46]). *DIO2*, *LPL*, *MAP1B*, *EDN3* were top ranked genes that explained the resulting vector field of young Hu-MuSCs with an increase in velocity. *NPTX2* and *KLF6* velocity were increased in cluster 1 and 7, clusters associated with adult and aged Hu-MuSCs (S3 Fig and S2 Table).

The gene ontology (GO) term analysis of differentially expressed genes in young, adult and aged Hu-MuSCs revealed an enrichment in ECM terms in young Hu-MuSCs although they were decreased in aged Hu-MuSCs. Type I / IFNy signaling were decreased in young Hu-MuSCs. Terms associated with muscle processes were increased in aged Hu-MuSCs and downregulated in adult Hu-MuSCs. An enrichment of cellular response to oxygen levels and hypoxia terms was also detected in aged Hu-MuSCs (Fig 4).

To summarize, our approach resulted in the identification of novel aging-related markers such as *CAV1* and *GREM1* while verifying in Hu-MuSCs the expression levels of genes that have been previously associated with murine aging alterations.

## scRNAseq analysis also reveals transcriptomic changes during aging in muscle progenitor cells, cycling Hu-MuSCs and myoblasts

Since our dataset also contained a large fraction of differentiated cells, we separately analyzed clusters encompassing the cycling Hu-MuSCs (cluster 8), muscle progenitors (cluster 4) and myoblasts (cluster 9) (Fig 5A). Top differentially expressed genes in the muscle progenitors included *MEST* (Mesoderm Specific Transcript, a negative regulator of adipocyte differentiation [47]), *HSPG2* (Heparan Sulfate Proteoglycan 2, encodes for secreted molecule perlecan, deposited on all basement membrane [48]), *OLFM2B* (Olfactomedin 2b, a regulator for TGF-b

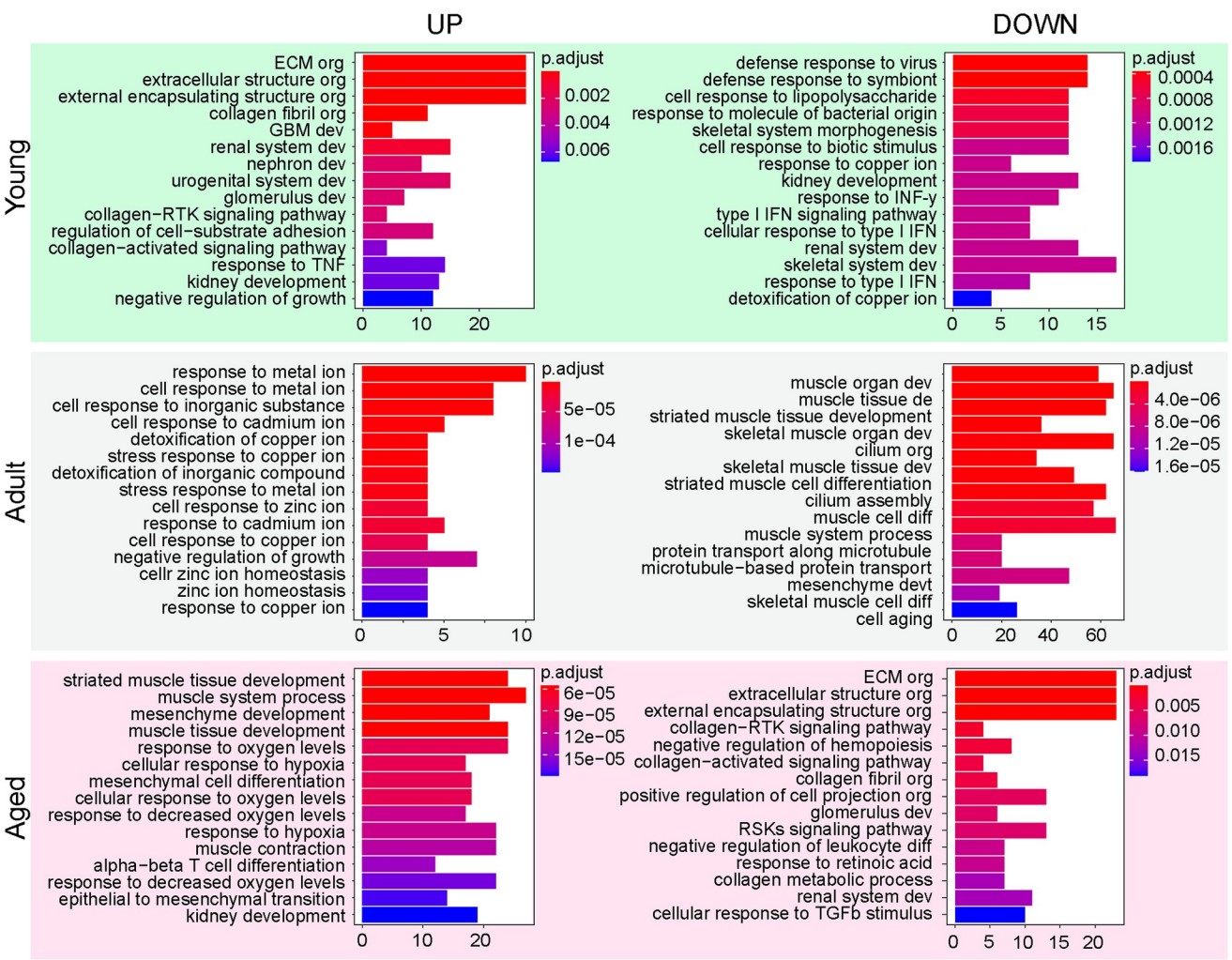

**Fig 4. Gene ontology enrichment upon aging in Hu-MuSCs.** Bar plots of gene ontology analysis of differentially up- and down- regulated genes in Hu-MuSCs for each age group.

[49]) and *SPARC* for the young samples; *FOS*, Metallothionein genes (*MT1E* and *MT2A*) and *SPARCL1* (SPARC-like protein 1, an ECM protein that has been described as a differentiation promotor of C2C12 cells [50]) for the adult samples; and *NUPR1* (Nuclear Protein 1 Transcription Regulator, a repressor of ferroptosis [51]), *TRDN* (Triadin, plays a role in muscle excitation-contraction [52]), *MTRNR2L12* (an isoform of humanin [53]) for the aged samples. As with our Hu-MuSC explorations, GO term analysis of differentially expressed genes for differentiated cell clusters showed an enrichment of ECM and cell matrix adhesion terms in young muscle progenitors, along with an enrichment of muscle cell differentiation and interferon gamma terms in aged muscle progenitor cells. Similar analysis for the cycling Hu-MuSCs revealed a significant increase of *TUBA1B* (Tubulin Apla-1B, a cytoskeleton protein [54]), *TYMS* (Thmidylate Synthetase, a critical enzyme for DNA replication and DNA repair [55]), *H2AFV* (H2A.Z Variant Histone 2), *STMN1* (Stathmin1, a microtubule-binding protein [56]) transcripts levels in young cells, similar to that of *SPARCL1*, *CXCL14*, *ZFP36* (ZFP36 Ring Finger Protein), *EIF1* (Eukaryote Translation Initiation Factor 1) in adult cells and Proteosome proteins (*PSMB10* and *PSMB9*), *PRDX1* (Peroxiredoxin 1, an antioxidant enzyme

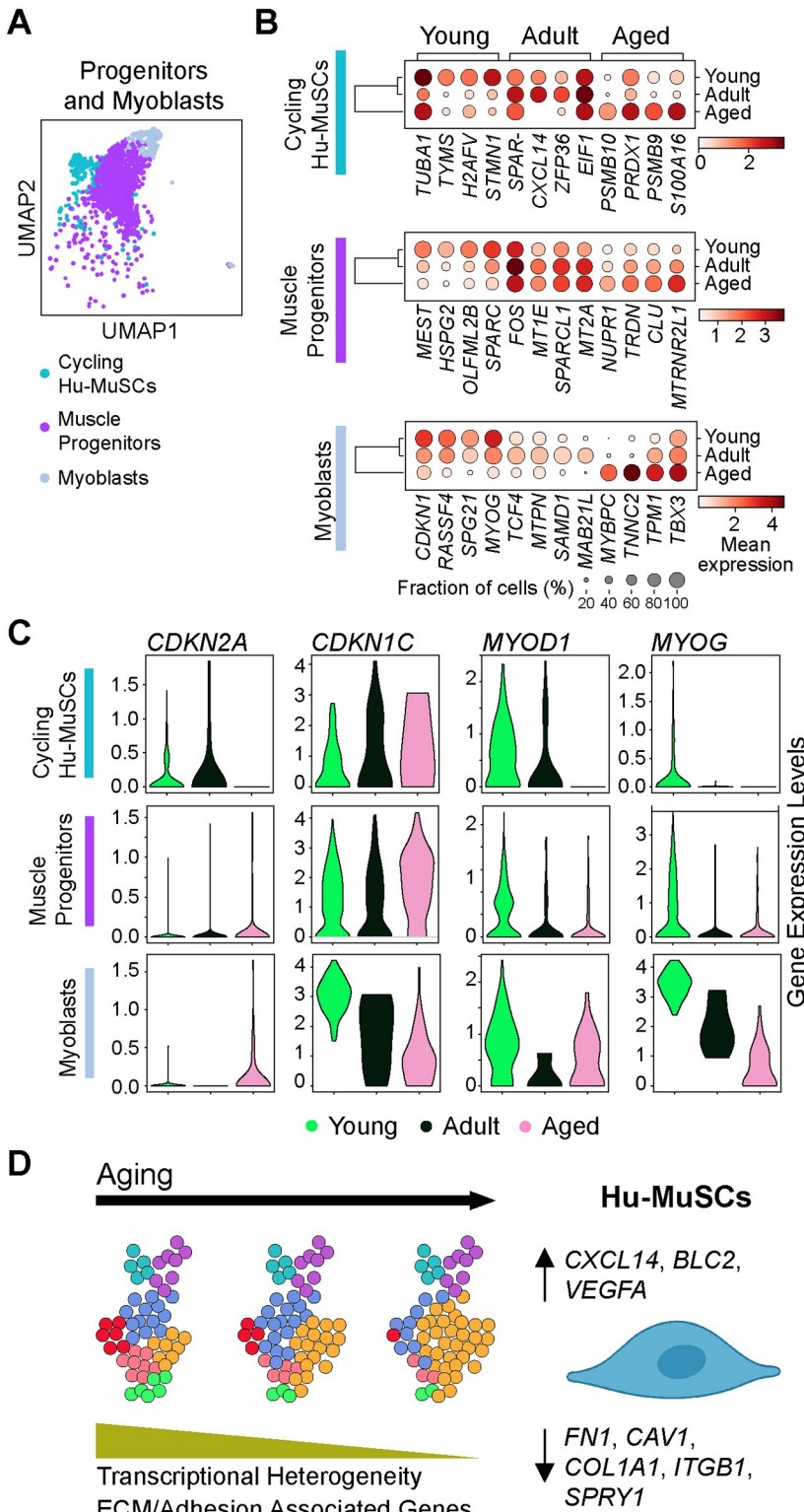

**Fig 5. Transcriptional analysis of progenitor and myoblast cells in the different age groups.** (A) UMAP of merged age groups for only muscle progenitors (cluster 4), cycling Hu-MuSCs (cluster 8) and myoblasts (cluster 9). (B) Dot plot displaying the top 4 genes differentially expressed for each age group in cluster 4, 8 and 9. (C) Violin plots of the expression of genes altered with aging. (D) Summary schematic of how Hu-MuSCs transcriptome changes with aging.

[57]) and *S100A16* (S100 Calcium Binding Protein A16) in aged cells. GO term analysis showed enrichment in DNA replication and cell cycle terms in young cycling Hu-MuSCs, cellular response to metal ion terms in adult cells and enrichment in antigen processing and Wnt signaling pathway in aged cycling Hu-MuSCs (S4 Fig). Finally, *CDKN1C*, *RASSF4* (Ras Association Domain Family Member 4), *SPG21* (SPG21 Abhydrolase Domain Containing, Maspardin, involved in repression of T cell activation [58]), *MYOG* (Myogenin) were significantly differentially expressed in young myoblasts, *TCF4* (Transcription Factor 4), *MTPN* (Myotrophin), *SAMD1* (Sterile Alpha Motif Domain Containing 1, an unmethylated CGI-binding protein [59]) and *MAB21L1* (Mab-21 Like 1, a putative nucleoidyltransferase [60]) in adult myoblasts and *MYBPC1* (Myosin Binding protein C1), *TNNC2* (Troponin C2, fast skeletal type), *TPM1* (Tropomysin) and *TBX3* (T-Box Transcription Factor 3) in aged myoblasts. Mitochondrial translation terms were enriched in young myoblasts while muscle processes, differentiation and development were enriched in aged myoblast (Fig 5B and S5 Fig).

In addition, we found that *CDKN2A* (Cyclin Dependent Kinase Inhibitor 2A) and *CDKN1C* expression increased with aging in cycling Hu-MuSCs and muscle progenitor cells. In contrast, MYOD1 and *MYOG* levels decreased with aging in those two clusters as well as in the myoblast cluster. *CDKN2A* transcription levels also increased in myoblasts upon aging while *CDKN1C* levels decreased in the myoblast cluster (Fig 5C).

Overall, these analyses provide new insights into transcriptomic modulations of more differentiated human muscle stem cells and muscle progenitors during aging summarily characterized by similar age-related GO term enrichment to Hu-MuSCs.

## Discussion

In this extension of our prior work, we performed extended transcriptomic analyses of isolated human muscle stem cells across a range of adult ages. We found that satellite cell aging is characterized by a global decrease in transcriptional heterogeneity. At the single gene level, new and previously described transcriptional age-related changes in human satellite cells were identified (Fig 5D).

The addition of new samples coupled with more complex computational analyses of our single cell RNA sequencing data allowed us to further understand the cellular heterogeneity of human satellite cells. While aging was not associated with the appearance or disappearance of age-specific clusters, we found that the cell distribution among the different clusters was altered. A loss of cellular heterogeneity during mouse muscle aging has been shown previously, Chakkalakal et al. found a decrease of labeling retaining-SCs (bone fide stem cells) while committed progenitors (non-labeling retaining-SCs) were preserved in aged mice [5]. This suggests that certain subpopulations of Hu-MuSCs are retained with aging while others are reduced or partially lost. Importantly, this loss of heterogeneity was associated with a decline in transplantation potential of aged SCs [5]. These prior studies together with this study suggest that a shift in satellite cell subpopulation representation may be responsible for impaired muscle regeneration in the aging population.

We found (1) the distribution of cells among the various human clusters we have characterized changes during aging and (2) genes including *CAV1*, *CXCL14*, *FN1* and *GPX3* that can explain this differential cell distribution. Our analysis highlighted gene-defining clusters that are significantly altered by aging. For example, we found that *CAV1*, a marker of a high transplantation potential Hu-MuSC subpopulation [21], decreases with age. We [21] and others [61] have previously described that *CAV1* is expressed in quiescent satellite cells with increased engraftment properties, although its functional role in aging satellite cells remains unknown. Indeed, CAV1 in aging is not well understood as opposing finding a have been described in

several tissues and studies [62–66]. Further functional studies in human SCs are necessary to determine the precise role of *CAV1* in aging and future transplantation studies will help determine if diminishment of this satellite cell subpopulation is responsible for decreased satellite cell function in aging.

CXCL14, transcripts of which we found increased in aging, has been shown to prevent cell cycle withdrawal and to be a negative regulator of myoblast differentiation. Experimental CXCL14 reduction ameliorates regenerative defects in aging mouse muscle [67]. We also found an increase of *GPX3* expression in aged Hu-MuSCs. GPX3 (glutathione peroxidase 3), a retinoid-responsive gene that mediates the antioxidant effects of retinoic acid in human myoblasts, may be important in muscle stem cell survival [37]. Therefore, our observed increase in *CXCL14* + and *GPX3+* Hu-MuSCs could be related to regenerative decline of aged human muscles.

Our in-depth computational analyses of human samples also corroborated other trends previously described in age-related mouse studies, notably a decrease in *ITGB1* and *SPRY1* expression in aging. Levels of ITGB1 and FAK (Focal Adhesion Kinase) were lower in aged satellite cells leading to decreased cell adhesion and increased cell death [13,14]. In addition to a decrease of ITGB1 and FAK, our analyses also reveal a decrease in ECM such as fibronectin and collagen associated genes. Early studies observed an increase of the external lamina materials surrounding the satellite cells in aged rodent muscles [68]. Fibronectin and collagens such as COL4 are constituents of the external lamina, excess of such trophic factors could be one explanation of the downregulation of those genes in Hu-MuSCs. Indeed, while extensive work has been done on ECM remodeling of the satellite cell's niche during aging [69], there is a lack of data describing the effect of aging on ECM components produced by satellite cells. Nevertheless, fibronectin and collagens produced by satellite cells are critical for the maintenance of their quiescence [70,71]. Our analyses suggest that aging may induce a decrease in ECM components expression by Hu-MuSCs which may have a role in ECM remodeling found in aging muscles [72].

SPRY1, a regulator of satellite cell return to quiescence [38] and detected in a subset of Hu-MuSCs in our previous [21] and present studies, also decreases in aged SCs. Comparable results were found in mice, where age-associated methylation suppression of SPRY1 leads to loss of the reserve stem cell pool [73] while SPRY1 over-expression in aged satellite cells in vivo preserves the SC pool [5]. Our findings add additional evidence supporting the concept that the high expressing *SPRY1* subset of SCs is critical for muscle regeneration during aging. We also found increased expression of other major drivers of ageing such as mTor pathway targets (e.i. *BCL2* and *VEGFA*) [74] being elevated in aged satellite cells. FOS was also found to be elevated in cluster 1 where most aged cells resided. Although, a recent study showed that Fos mRNA is a feature of freshly isolated satellite cells from uninjured muscle and that it marks a subset of satellite cells with enhanced regenerative ability [75], how FOS levels impact aging in human muscle stem cells still remains to be fully elucidated. Finally, our dataset also captured more differentiated muscle stem cells in which *CDKN2A* expression level was increased and *MYOD1* and *MYOG* levels were decreased with age. Indeed, increased level of *CDKN2A* has been described in geriatric human and mouse muscle stem cells to induce a loss of reversible quiescence, a pre-senescence state and result in failure to proliferate and differentiate [6]. Although we cannot totally exclude the role different muscle type in Hu-MuSCs heterogeneity, altogether, with a limited number of samples, our human satellite cell transcriptomic study was able to validate age-related mouse findings, confirm the potential role of age-related pathways in Hu-MuSCs during aging, and identify changes in satellite cell distribution among subpopulations.

Our RNA velocity analysis identified additional genes with age-specific differential velocity which explained the vector field of the different Hu-MuSCs age groups. While some of those will need further experimental investigation to understand their mechanism of action during

aging (e.g *CDKN1C*, *DIO2*, *KLF6*, *NPTX2*, *MAP1B*), other genes, for which rodent experiments have been carried out in homeostasis, may explain the loss of satellite cell stemness and self-renewal such as *GREM1* and *EDN3*. *GREM1* velocity was associated with aged Hu-MuSCs. A BMP antagonist, GREM1 would be expected to induce a decrease in satellite cell number [40] possibly by acting as a negative regulator of satellite cell self-renewal. Therefore GREM1 could account for the loss of self-renewal and reduced number of satellite cells in aged human muscle. Separately, we found that *EDN3* explains the vector field of young Hu-MuSCs. *EDN3* is expressed in quiescent mouse satellite cells [76] and is down-regulated as they become activated [77,78] suggesting that EDN3 could play a functional role and/or be a new marker for the loss of quiescence with aging.

This study is the first report of single cell transcriptomes of human satellite cells at various stages of aging. The possibility exists that our representation of aging human satellite cell transcriptomes is incomplete with a limited number of samples. However, since we were able to confirm previous mouse observations, it is likely that this study does contain a faithful and adequate sampling of human muscles to describe the major alterations of the transcriptomic landscape in aging. In situ validations of gene expression (*CAV1*, *SPRY1*, *ITGB1* and *PAX7*) and protein, as well as functional studies will further elucidate the roles of the different genes identified here.

## Supporting information

**S1 Fig. BBKNN sorted human muscle stem cells for each age group.** (A) As shown in Fig 1A, samples were first merged into their own age group. UMAP displaying clusters and samples for each age group. (B) Dot plots displaying the expression of myogenic, cycling, stemness and cluster marker genes for each cluster in each age group. (C) Dot plots displaying the expression levels of an identical gene set in the young, adult and aged Hu-MuSCs.
(TIF)

**S2 Fig. INGEST merging of sorted human muscle stem cells.** (A) UMAP of the INGEST analysis displaying all 12 samples. (B) MAP of each age group and their distribution in clusters. (C) Proportion plot of cells assigned to each cluster for each age group.
(TIF)

**S3 Fig. Expression and velocity of relevant genes that inferred age-specific differential velocity.**
(TIF)

**S4 Fig. Cycling Hu-MuSCs GO term analysis upon aging.** Bar plots of gene ontology analysis of differentially up-regulated genes in the cycling Hu-MuSCs cluster (8) for each age group.
(TIF)

**S5 Fig. Progenitor and myoblast cells GO term analysis upon aging.** Bar plots of gene ontology analysis of differentially up-regulated genes in the muscle progenitor cluster (4), and myoblasts cluster (9) for each age group.
(TIF)

**S1 Table. Demographics and sample characteristics collected and used for downstream analysis.**
(TIF)

**S2 Table. Gene ranking for each age group resulting from differential velocity t-test.**
(CSV)

## Acknowledgments

The authors would like to thank all the organ and tissue donors and their families for their generous donation.

## Author Contributions

**Conceptualization:** Emilie Barruet, Pauline Marangoni, Jason H. Pomerantz.

**Data curation:** Emilie Barruet.

**Formal analysis:** Emilie Barruet.

**Investigation:** Emilie Barruet, Katharine Striedinger.

**Methodology:** Emilie Barruet.

**Resources:** Jason H. Pomerantz.

**Software:** Emilie Barruet.

**Supervision:** Jason H. Pomerantz.

**Validation:** Emilie Barruet.

**Visualization:** Emilie Barruet.

**Writing – original draft:** Emilie Barruet.

**Writing – review & editing:** Emilie Barruet, Pauline Marangoni, Jason H. Pomerantz.

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
