## [Decision Letter · Decision Letter 0]

29 Nov 2022

PONE-D-22-29728Loss of transcriptional heterogeneity in aged human muscle stem cellsPLOS ONE

Dear Dr. Pomerantz,

Thank you for submitting your manuscript to PLOS ONE. After careful consideration, we feel that it has merit but does not fully meet PLOS ONE’s publication criteria as it currently stands. Therefore, we invite you to submit a revised version of the manuscript that addresses the points raised during the review process.

We look forward to receiving your revised manuscript.

Kind regards,

Atsushi Asakura, Ph.D

Academic Editor

PLOS ONE

“This work was supported by NIH R01AR072638-03 to JHP. The authors would like to thank all the organ and tissue donors and their families for their generous donation.”

“This work was supported by NIH R01AR072638-03 to JHP

Reviewers' comments:

Reviewer's Responses to Questions

**Comments to the Author**

1. Is the manuscript technically sound, and do the data support the conclusions?

Reviewer #1: Yes

Reviewer #2: Yes

2. Has the statistical analysis been performed appropriately and rigorously? 

Reviewer #1: N/A

Reviewer #2: Yes

3. Have the authors made all data underlying the findings in their manuscript fully available?

Reviewer #1: Yes

Reviewer #2: Yes

4. Is the manuscript presented in an intelligible fashion and written in standard English?

Reviewer #1: Yes

Reviewer #2: Yes

5. Review Comments to the Author

Reviewer #1: This group previously reported a pioneering study on the heterogeneity of human muscle satellite cells by scRNA-seq analyses. This study made some progress by comparing young, adult, and aged muscle satellite cells. First, the authors found the age-related loss of global transcriptomic heterogeneity in the scRNA-seq analyses. In addition, the authors identified new markers, CAV1, CXCL14, and GPX3 that are altered during aging in human satellite cells. As our understanding of satellite cell aging at the single-cell level is lacking, this study will provide valuable information for us. Please respond to the following concerns.

1. Pectoralis major muscle was not included in the aged muscle. Does not the composition of different muscles affect the heterogeneity of muscle satellite cells?

2. Figure 1B; Why does the mesenchymal cell cluster (Cluster 12) get positioned closer to MuSCs than myocytes (Cluster 11) in the UMAP analysis?

3. S.Figure 1: Please indicate which cluster of young MuSC corresponds to adult or aged MuSC cluster numbers. In addition, different genes were used to classify the cells among young, adult, and aged MuSCs. It is easier for readers to understand if the same gene set is used?

4. Figure 2B; Increased myocyte cluster is a characteristic of aged MuSCs. If this observation is true, MYOD or MYOG-positive cells are frequently detected in aged MuSCs. The authors should confirm these results using IHC.

5. Basically, all presented data are from dry experiments. To confirm the data of scRNA-seq analyses, at least immunostaining of CAV1, SPRY1, and ITGB1 with PAX7 should be performed in both young and aged MuSCs.

6. Reviewer recommends including the following paper in the discussion of ECM organization. Do the authors observe the appearance of the external lamina in the satellite cell-myofiber interspace? Are there genes involved in the appearance of the external lamina between satellite cell and myofiber?

Cell Tissue Res. 1977 Dec 19;185(3):399-408. doi: 10.1007/BF00220299.

Reviewer #2: Comments to Authors

This work by Barruet et al. explored the transcriptomic changes in human satellite cells with age using single cell RNA-seq. The authors showed age-related loss of heterogeneity and identified several genes that are altered during aging in human satellite cells.

This report provides a new interesting perspective on the regulation of human satellite cell during aging. However, there are several points which require clarifications before acceptance for publication.

Specific comments

1)In line 85 which stated downsampling was performed, were the authors able to determine that downsampling did not affect the detection of significant changes between groups that could be due to effect size? How was this lack of effects determined?

2)In line 68 which stated sample selection, were there efforts to exclude any underlying pathology involving the selected muscle samples or exclude familiar/ hereditary or sporadic PNS or CNS diseases that the patient may have which may influence the transcriptome? Please describe briefly how this was excluded.

3)In line 151, the authors mentioned about MX1 positive satellite cells which have been described by Scaramozza et al (Cell Stem Cell, 2019). Scaramozza and colleagues reported those satellite cell subpopulation expressed high level of Pax3, and showed stress tolerance. However, in this report, cluster 10 satellite cells showed low level of Pax3. Therefore, those satellite cell subpopulation seems different from the one previously reported. The authors should reconsider this subpopulation.

6. PLOS authors have the option to publish the peer review history of their article (what does this mean?). If published, this will include your full peer review and any attached files.

Reviewer #1: **Yes: **So-ichiro Fukada

Reviewer #2: No

---

## [Author Response · Author response to Decision Letter 0]

15 Mar 2023

Dear reviewers and editors,

I am pleased to re-submit our manuscript entitled “Loss of transcriptional heterogeneity in aged human muscle stem cells” (PONE-D-22-29728) that I hope will now find ready for publication in PLOS ONE. We have addressed each of the academic editor and reviewers’ points with additional data analysis and revisions to the text which were detailed below. The revised manuscript is strengthened with additional discussion, figure panel and clarification.

Should our manuscript be accepted for publication we will provide repository access to our single cell gene expression fastq files and filtered matrices which have been deposited onto the Gene Expression Omnibus repository (GSE196554). Additionally, our detailed scripts for our analysis will become publicly accessible here https://github.com/EmilieB12/Aging_Hu-MuSCs.

Thank you very much for considering our paper. I have included our detailed point-by-point response to the reviewer’s and editor’s comments below. Please be aware that the full version of the response to reviewers letter (including supporting figures) can be found as a standalone file uploaded alongside the article files given that the text boxes do not allow for figures to be shared

Sincerely,

Jason Pomerantz, MD

Reviewer #1: This group previously reported a pioneering study on the heterogeneity of human muscle satellite cells by scRNA-seq analyses. This study made some progress by comparing young, adult, and aged muscle satellite cells. First, the authors found the age-related loss of global transcriptomic heterogeneity in the scRNA-seq analyses. In addition, the authors identified new markers, CAV1, CXCL14, and GPX3 that are altered during aging in human satellite cells. As our understanding of satellite cell aging at the single-cell level is lacking, this study will provide valuable information for us. Please respond to the following concerns.

1. Pectoralis major muscle was not included in the aged muscle. Does not the composition of different muscles affect the heterogeneity of muscle satellite cells?

Thank you for raising this point. We have previously shown that numerous cluster-specific markers are conserved across various muscles including vasti lateralis, recti femoris, recti abdominis and pectoralis major (Barruet et al. Elife. 2020). In addition, we have confirmed that major markers of muscle stem cell aging which have been extensively studied in mice are found in our human datasets as well. Altogether, these findings suggest that the skeletal muscle type does not impact the heterogeneity of muscle satellite cells in terms of the characterization we undertook. However, we cannot formally exclude that muscle type may modulate the transcriptional heterogeneity of muscle satellite cells, and thus we added a statement in our discussion (line 366-367).

2. Figure 1B; Why does the mesenchymal cell cluster (Cluster 12) get positioned closer to MuSCs than myocytes (Cluster 11) in the UMAP analysis?

Single cell analyses use unsupervised clustering as an unbiased way to identify various cell populations. Although cluster 11 and cluster 12 were positioned based on their transcriptional profile relative to each other and to all other clusters, it does not suggest degree of phenotypic difference of these two cell populations. 

3. S.Figure 1: Please indicate which cluster of young MuSC corresponds to adult or aged MuSC cluster numbers. In addition, different genes were used to classify the cells among young, adult, and aged MuSCs. It is easier for readers to understand if the same gene set is used?

Thank you for the suggestion. Accordingly, we added panel C in S Fig1, which showcases the expression levels of an identical gene set in the young, adult and aged Hu-MuSCs to enable our readers to compare unbiased marker expression dotplots (panel B) and selected marker expression dotplots (panel C) to understand how relatable clusters are between the three age groups. Both were generated using the same resolution (0.5). 

4. Figure 2B; Increased myocyte cluster is a characteristic of aged MuSCs. If this observation is true, MYOD or MYOG-positive cells are frequently detected in aged MuSCs. 5. The authors should confirm these results using IHCBasically, all presented data are from dry experiments. To confirm the data of scRNA-seq analyses, at least immunostaining of CAV1, SPRY1, and ITGB1 with PAX7 should be performed in both young and aged MuSCs.

We recognize the importance of analyzing protein expression, however we feel that it is beyond the scope of this study. We have tried, but as the reviewer is aware, analyzing differences in protein levels in satellite cells by immunostaining is challenging and not always feasible with available antibodies. The inconsistencies are further complicated by the fact that many proteins that we would analyze exist in various cellular compartments (e.g. CAV1) that would also affect quantification. Since the findings of additional staining would not change the conclusions of our manuscript, we have not included new experiments here. However, we did modify the text acknowledging the future goals of understanding protein expression and mechanistic analysis of how phenotype is affected on line 389-390.

6. Reviewer recommends including the following paper in the discussion of ECM organization. Do the authors observe the appearance of the external lamina in the satellite cell-myofiber interspace? Are there genes involved in the appearance of the external lamina between satellite cell and myofiber?

Cell Tissue Res. 1977 Dec 19;185(3):399-408. doi: 10.1007/BF00220299.

Thank you for this recommendation. We have reviewed the literature further, and now reference the suggested paper in the discussion section of our manuscript (line 342-345) and discuss the potential involvement of genes we found altered in our study in increased external lamina between satellite cell and myofibers.

Reviewer #2: Comments to Authors

This work by Barruet et al. explored the transcriptomic changes in human satellite cells with age using single cell RNA-seq. The authors showed age-related loss of heterogeneity and identified several genes that are altered during aging in human satellite cells.

This report provides a new interesting perspective on the regulation of human satellite cell during aging. However, there are several points which require clarifications before acceptance for publication.

Specific comments

1)In line 85 which stated downsampling was performed, were the authors able to determine that downsampling did not affect the detection of significant changes between groups that could be due to effect size? How was this lack of effects determined?

Thank you for raising this crucial point. Since our adult aged group had 6 times more cells than the other two groups, in order to avoid cofounder effects resulting from imbalanced cell number in age groups, we performed downsampling which is a state-of-art approach to eliminate such bias (Bhaduri et al, BMC Biology, 2018). That study is referenced, and the point explained in line 86. Notably, the process was verified to preserve clusters in order not to induce a bias.

2)In line 68 which stated sample selection, were there efforts to exclude any underlying pathology involving the selected muscle samples or exclude familiar/ hereditary or sporadic PNS or CNS diseases that the patient may have which may influence the transcriptome? Please describe briefly how this was excluded.

All muscle specimens were collected solely from patients with healthy muscle, who were undergoing surgery for other reasons unrelated to muscle injury or disease. Subjects with hereditary or sporadic PNS or CNS diseases that could affect muscle were excluded from our study. Criteria for inclusion are described in the Methods section.

3)In line 151, the authors mentioned about MX1 positive satellite cells which have been described by Scaramozza et al (Cell Stem Cell, 2019). Scaramozza and colleagues reported those satellite cell subpopulation expressed high level of Pax3, and showed stress tolerance. However, in this report, cluster 10 satellite cells showed low level of Pax3. Therefore, those satellite cell subpopulation seems different from the one previously reported. The authors should reconsider this subpopulation.

In response to the Reviewer’s comment, we now discuss the Pax3 population in lines 157-159. The low levels of Pax3 transcripts likely reflects known limitations of single cell RNA sequencing sensitivity as well as the rarity of the population. In addition, Pax3 has mostly been studied in mice or during development. Its role and expression pattern in adult human muscle stem cell has yet to be fully elucidated (Mierzejewski, seminars in cell & dev biol, 2020).

---

## [Decision Letter · Decision Letter 1]

13 Apr 2023

Loss of transcriptional heterogeneity in aged human muscle stem cells

PONE-D-22-29728R1

Dear Dr. Pomerantz,

We’re pleased to inform you that your manuscript has been judged scientifically suitable for publication and will be formally accepted for publication once it meets all outstanding technical requirements.

Kind regards,

Atsushi Asakura, Ph.D

Academic Editor

PLOS ONE

Additional Editor Comments (optional):

Reviewers' comments:

Reviewer's Responses to Questions

**Comments to the Author**

1. If the authors have adequately addressed your comments raised in a previous round of review and you feel that this manuscript is now acceptable for publication, you may indicate that here to bypass the “Comments to the Author” section, enter your conflict of interest statement in the “Confidential to Editor” section, and submit your "Accept" recommendation.

Reviewer #1: All comments have been addressed

Reviewer #2: All comments have been addressed

2. Is the manuscript technically sound, and do the data support the conclusions?

Reviewer #1: Yes

Reviewer #2: (No Response)

3. Has the statistical analysis been performed appropriately and rigorously? 

Reviewer #1: Yes

Reviewer #2: (No Response)

4. Have the authors made all data underlying the findings in their manuscript fully available?

Reviewer #1: Yes

Reviewer #2: (No Response)

5. Is the manuscript presented in an intelligible fashion and written in standard English?

Reviewer #1: Yes

Reviewer #2: (No Response)

6. Review Comments to the Author

Reviewer #1: The authors sincerely considered all comments raised by this reviewer and addressed all concerns. Therefore, this reviewer recommends to publish this work in PLoS One.

Reviewer #2: (No Response)

7. PLOS authors have the option to publish the peer review history of their article (what does this mean?). If published, this will include your full peer review and any attached files.

Reviewer #1: **Yes: **So-ichiro Fukada

Reviewer #2: No

---

## [Editor Report · Acceptance letter]

3 May 2023

PONE-D-22-29728R1 

Loss of transcriptional heterogeneity in aged human muscle stem cells 

Dear Dr. Pomerantz:

I'm pleased to inform you that your manuscript has been deemed suitable for publication in PLOS ONE. Congratulations! Your manuscript is now with our production department. 

Kind regards, 

on behalf of

Dr. Atsushi Asakura 

Academic Editor

PLOS ONE